Two new temporary ectoparasitic isopods (Cymothoida: Cymothooidea) from Korean waters with a note on geographical distributions of Rocinela Leach, 1818 and Gnathia Leach, 1814

Kim Sung Hoon 1
Kim Jong Guk 2
Yoon Seong Myeong 3 4 smyun@chosun.ac.kr
1 Division of Ocean Sciences, Korea Polar Research Institute , Incheon , South Korea
2 Division of Zoology, Honam National Institute of Biological Resources , Mokpo , South Korea
3 Educational Research Group for Age-associated Disorder Control Technology, Graduate School, Chosun University , Gwangju , South Korea
4 Department of Biology, College of Natural Sciences, Chosun University , Gwangju , South Korea
Semprucci Federica
Electronic publication date: 2023 Jan 3
Publication date: 2023
Volume: 11
Electronic Location ID: e14593
Received 2022 Aug 29; Accepted 2022 Nov 28
Copyright: © 2023 Kim et al.
Copyright year: 2023
Copyright holder: Kim et al.
License: This is an open access article distributed under the terms of the Creative Commons Attribution License, which permits unrestricted use, distribution, reproduction and adaptation in any medium and for any purpose provided that it is properly attributed. For attribution, the original author(s), title, publication source (PeerJ) and either DOI or URL of the article must be cited.
License URL: https://creativecommons.org/licenses/by/4.0/

Keywords: Ectoparasite, Gnathia, Isopods, Morphology, Rocinela, Taxonomy, South Korea

Funding: Chosun University 2022 National Institute of Biological Resources (NIBR) 201902204 This study was supported by research funds from Chosun University (2022) and a grant (201902204) of the National Institute of Biological Resources (NIBR) funded by the Ministry of Environment (MOE), the Republic of Korea. The funders had no role in study design, data collection and analysis, decision to publish, or preparation of the manuscript.

==============================
Two new species of temporary ectoparasitic isopods, Rocinela excavata sp. nov. and Gnathia obtusispina sp. nov., are reported from the southern Islands of the Korean Peninsula. Rocinela excavata sp. nov. is distinguishable from its related species by the following characteristics: (1) laterally stepped rostrum; (2) separated eyes; (3) propodal blade having eight robust setae; and (4) merus having four or five blunt robust setae in pereopods 1–3. Gnathia obtusispina sp. nov. differs from its congeners by the combination of the following characteristics: (1) body covered with numerous tubercles and setae, (2) cephalon having tooth-like paraocular ornamentations; and (3) frontal border having two inferior frontolateral processes. These two new species are the 13th Rocinela species and 19th Gnathia species in the temperate Northern Pacific region, respectively. Discovery of these new species represents high species diversity of the genera Rocinela Leach, 1818 and Gnathia Leach, 1814 worldwide as well as in the Northern Pacific region. In addition, faunal diversity analysis on the members of both genera revealed that Rocinela species show high-latitude diversity whereas Gnathia species have low-latitude diversity.

Introduction

Within isopod taxa, the superfamily Cymothooidea Leach, 1814 including families Aegidae White, 1850 and Gnathiidae Leach, 1814 is predominantly parasites of fish or other crustaceans (Williams & Boyko, 2012; Smit, Bruce & Hadfield, 2019). Among the Cymothooideans, both Aegidae and Gnathiidae are known to be temporary ectoparasites that can attach to fishes (Bruce, 2009; Svavarsson & Bruce, 2012; Williams & Boyko, 2012; Cardoso et al., 2017; Smit, Bruce & Hadfield, 2019). However, aegids are also regarded as free-living micro-predators because they often detach from their hosts and spend most of their time free-living on the seafloor (Bruce, 2009; Williams & Boyko, 2012; Smit, Bruce & Hadfield, 2019). They morphologically differ from other Cymothooideans in terms of the maxillule having robust setae distally, maxillipedal palp articles 3 and 4 having conspicuous recurved robust setae distally, prehensile pereopods 1–3, and ambulatory pereopods 4–7 (Bruce, 2009). Similarly, adults of gnathiid isopods are also free-living (non-feeding) on cryptic habitats of sponges, dead corals, barnacle nests, and polychaete’s tube (Kopuz et al., 2011) whereas their juveniles show a hematophagous life cycle (Svavarsson & Bruce, 2012; Williams & Boyko, 2012; Smit, Bruce & Hadfield, 2019). Although gnathiids show highly polymorphic forms depending on their developmental stages and their adults exhibit considerable sexual dimorphism, they are distinguishable from other cymothooideans largely based on the adult male’s characteristics of having remarkably enlarged mandibles and only five pairs of pereopods (Svavarsson & Bruce, 2012; Ota, 2014; Boxshall & Hayes, 2019; Smit, Bruce & Hadfield, 2019).

So far, seven Rocinela species have been recorded from the Far East where the survey region of the present study is located: Rocinela belliceps (Stimpson, 1864) from the Sea of Okhotsk, Russia; Rocinela maculata Schioedte & Meinert, 1879 from the East Sea, Russia and South Korea; Rocinela japonica Richardson, 1898 from the Hakodate Bay, Japan; Rocinela affinis Richardson, 1904 from the Shizuoka, Japan; Rocinela angustata Richardson, 1904 from the Manazuru, Japan; Rocinela niponia Richardson, 1909 from the Sado Island, Japan and Chujado Island, South Korea; and Rocinela lukini Vasina, 1993 from the Sea of Okhotsk, Russia (Schioedte & Meinert, 1879; Richardson, 1898, 1904, 1909; Vasina, 1993; National Institute of Biological Resources, 2012; Kim & Yoon, 2020). Eleven Gnathia species have been reported in the Far East: Gnathia tuberculata Richardson, 1909 from Nanao, Japan; Gnathia derzhavini Gurjanova, 1933 from Askold Island, Russia; Gnathia rectifrons Gurjanova, 1933 from the East Sea, Russia; Gnathia schmidti Gurjanova, 1933 from the Bay of Vladimir, Russia; Gnathia bungoensis Nunomura, 1982 from the Saeki Bay, Japan; Gnathia nasuta Nunomura, 1992 from Kumamoto and Okinawa Islands, Japan; Gnathia sanrikuensis Nunomura, 1998 from the Otsuchi Bay, Japan; Gnathia capillata Nunomura & Honma, 2004 from Sado Island, Japan; Gnathia mutsuensis Nunomura, 2004 from Asamushi, Japan; Gnathia gurjanovae Golovan, 2006 from Peter the Great Bay, Russia; and Gnathia koreana Song & Min, 2018 from Geomundo Island, South Korea (Boyko et al., 2008; Song & Min, 2018; Shodipo et al., 2021).

In this study, we report two temporary ectoparasitic isopods from Korean waters with their detailed descriptions and illustrations. Geographical distributions of these two genera are also discussed.

Materials and Methods

All materials were collected at the bottom of sublittoral zones using a Smith-McIntyre grab and SCUBA diving. Rocinela specimens were sampled from sandy-mud flats by using the Smith-McIntyre grab. Gnathia specimens were collected from the bryozoans and seaweeds on bedrock. SCUBA diving was used to survey the bedrock of sublittoral zones. These collected materials were immediately fixed in 95% ethyl alcohol and then transferred to the laboratory. Isopods were sorted from the transferred materials and then observed and dissected under a dissecting microscope (Olympus SZH-ILLD, Japan). Measurements and drawings of specimens were conducted with the aid of a drawing tube on a compound microscope (Olympus, BX50, Shinjuku, Tokyo, Japan) or the dissecting microscope. Pencil drawings were digitally scanned, inked, and arranged using a tablet and Adobe Illustrator CS6 as mentioned in Coleman (2003, 2009). All examined type series and additional material were moved into each small glass vial filled with 95% ethanol and deposited at the National Institute of Biological Resource (NIBR), South Korea.

The electronic version of this article in Portable Document Format (PDF) will represent a published work according to the International Commission on Zoological Nomenclature (ICZN), and hence the new names contained in the electronic version are effectively published under that Code from the electronic edition alone. This published work and the nomenclatural acts it contains have been registered in ZooBank, the online registration system for the ICZN. The ZooBank LSIDs (Life Science Identifiers) can be resolved and the associated information viewed through any standard web browser by appending the LSID to the prefix http://zoobank.org/. The LSID for this publication is: (urn:lsid:zoobank.org:pub:7A53937A-F2EB-49C7-B8DA-F0AA36241310). The online version of this work is archived and available from the following digital repositories: PeerJ, PubMed Central and CLOCKSS.

Results

Taxonomy

Order Isopoda Latreille, 1817

Suborder Cymthoida Wägele, 1989

Superfamily Cymothooidea Leach, 1814

Family Aegidae White, 1850

Genus Rocinela Leach, 1818

Type species. Rocinela danmoniensis Leach, 1818 by monotypy.

Diagnosis. Body typically flat, slightly vaulted dorsally; rostrum blunt, covering all or part of antennular peduncles; eyes large, sometimes fused each other, occupying over 50% width of the cephalon; pleonite 1 not abruptly narrower than pereonite 7; antennule shorter than antenna, with distinct peduncles; mandibular incisor narrow, not divided and denticulate; maxillipedal palp consisting of three articles; maxillipedal endite present; pereopod 1–3 with robust setae on propodus; pleopodal endopods 3–4 without plumose setae marginally; uropodal protopod mesially produce; uropodal rami lamellar; pleoptelson distally rounded (Brusca & France, 1992; Bruce, 2009).

Remarks. Synonymy and diagnosis have been well recognized by Brusca & France (1992) and Bruce (2009) and we followed them in this study. Among the members of this family, the genus Rocinela Leach, 1818 is distinguishable from other genera by having pleonite 1 not abruptly narrowing than pereonite 7 and a three-articled maxillipedal palp (Bruce, 2009). Although Rocinela species show a quite uniform appearances to each other, the shape of the frontal margin of the cephalon and the pereopodal armature are most helpful in identifying species (Brusca & France, 1992; Bruce, 2009). Rocinela signata Schioedte & Meinert, 1879 is one of the rare isopods that is known to attack humans (Garzón-Ferreira, 1990).

Rocinela excavata sp.nov.

urn:lsid:zoobank.org:act:9A4CC86D-6930-4FC6-9FC9-DBF105A2B285

Figures 1–3

Figure 1 Rocinela excavata sp. nov., holotype, male.

(A) Habitus, dorsal view; (B) Habitus, lateral view; (C) Cephalon, dorsal view; (D) Distal end of cephalon, ventral view; (E) Distal end of pleotelson; (F) Antennule; (G) Antenna; (H) Mandible; (I) Serrate seta of mandibular palp; (J) mandibular incisor; (K) Maxillule; (L) Distal end of maxillule; (M) Maxilla; (N) Distal end of maxilla; (O) Maxilliped; (P) Distal end of maxilliped. Scale bars: A, B = 5 mm, C–E = 2 mm, F, G = 1 mm; H, K, M, O = 0.5 mm, N = 0.2 mm, L, P = 0.1 mm, I, J = 0.05 mm.

Figure 2 Rocinela excavata sp. nov., holotype, male.

(A) Pereopod 1; (B) Propodal blade of pereopod 1; (C) Pereopod 2; (D) Robust seta of propodal bladed in pereopod 2; (E) Pereopod 3; (F) Pereopod 4; (G) Pereopod 5; (H) Pereopod 6; (I) Pereopod 7. Scale bars: A, C, E, F–I = 1 mm, B = 0.1 mm; D = 0.05 mm.

Figure 3 Rocinela excavata sp. nov., holotype, male.

(A) Pleopod 1; (B) Pleopod 2; (C) Pleopod 3; (D) Pleopod 4; (E) Pleopod 5; (F) Uropod. Scale bar: A–F = 1 mm.

Type material.—Holotype, designated here: South Korea: ♂, 19.3 mm, Chujado Island (33°58′50″N, 126°20′23″E), Chuja-myeon, Jeju-si, Jeju-do, 15 January 2019, 30–40 m, gravelly mud flats, S.H. Kim leg., Smith-McIntyre grab, NIBRIV0000900845. Paratype: 1♂, the same location as holotype, NIBRIV0000895341.

Description of holotype male. Body (Figs. 1A and 1B), 2.1 times longer than width, oval, dorsoventrally depressed; dorsal surface smooth. Cephalon (Figs. 1C and 1D) triangular; posterior margin slightly tri-sinuated, but not distinct; rostrum truncated anteriorly, stepped laterally; eyes large, separate. Pereonite 1 slightly longer than other pereonites; pereonite 3 widest; pereonite 7 narrower than preceding pereonites, tapering posteriorly. Coxal plates visible on dorsal side, acute posteriorly; coxal furrows present in coxal plates 4–7. Pleonite 1 hidden by pereonite 7, slightly visible on both lateral sides; pleonites 2–4 with subacute apex, but pleonite 5 with blunt apex. Pleotelson (Fig. 1E) semicircle or shield-shaped, tapering posteriorly, with numerous plumose setae and robust setae distally; lateral margins concave proximally; dorsal surface with one pair of depressions proximally and one medial carina.

Antennule (Fig. 1F) reaching anterior margin of pereonite 1; peduncular article 1 wider than article 2, with two penicillate setae distally; article 2 subequal to article 1 in length, with three penicillate setae and one simple seta laterally; article 3 elongated oblong, longest, 1.7 times longer than article 2, with one penicillate seta and two short simple setae distally; flagellar article 1 rectangular, 0.3 times as long as peduncular article 3, without setae; articles 2–5 square, with two aesthetascs distally; article 6 min, with two aesthetascs, one penicillate seta, and three simple setae. Antenna (Fig. 1G) exceeding beyond posterior margin of pereonite 1; peduncular article 1 globular; article 2 short, with three simple setae distally; article 3 4.7 times longer than article 2, with one simple seta distally; article 4 oblong, 1.5 times longer than article 3, with one simple seta; article 5 elongated, longest, 1.3 times longer than article 4, with three penicillate setae and three simple setae distally; flagellum consisting of 16 articles; each article with short simple setae distally except for first article without setae.

Frontal lamina (Fig. 1D) short, subacute distally; labrum projecting downwardly. Mandible (Figs. 1H–1J), incisor acute, with one process covered by minute spinous papulae; molar process rounded; palp article 2 longer than others, with 10 serrated setae and two long simple setae along with lateral margin; article 3 with 17 serrate setae (bifurcated distally) laterally. Maxillule (Figs. 1K and 1L) slender, four robust setae distally; apex acute. Maxilla (Figs. 1M and 1N) stout proximally; inner lobe with one curved robust seta distally and two protrusions laterally; outer lobe with two curved robust setae. Maxilliped (Figs. 1O and 1P), first article oblong, 2.6 times longer than width, wider posteriorly; second article 0.3 times as long as first article, with one curved robust seta and long simple seta distally; third article 0.6 times longer than second article, with four curved robust setae and one simple seta distally.

Pereopods 1–3 (Figs. 2A–2E), basis oblong, with 1–4 penicillate setae on superior margin and one simple seta at inferodistal angle; ischium almost 0.5 times as long as basis, expanding superior distal end, with one or two robust setae superodistally; merus trapezoidal, with several robust and simple setae at superodistal angle and four blunt robust setae along with inferior margin, but pereopod 2 with five blunt robust setae; carpus shortest, about 0.3 times as long as merus, with one robust seta on inferodistal end; propodus almost three times longer than carpus, with blade on palm; propodal blade 0.7 times as long as wide, with eight robust setae distally and one long simple seta proximally; robust setae with one simple setule distally; dactylus curved, as long as propodus, without setae. Pereopods 4–7 (Figs. 2F–2I), articles sequentially shortened; basis with 3–6 penicillate setae superiorly and two robust setae inferodistally, longest; ischium to carpus with tubercles and robust setae along with inferior margins, and robust setae at superior distal angles; propodus with several tubercles and robust setae along with inferior margin, and one penicillate seta and several simple setae at superior distal angle; dactylus slightly curved, without setae.

Pleopods (Figs. 3A–3E) sequentially larger posteriorly; pleopods 2–4 with globular patterns along with lateral margins of exopods. Pleopod 1 (Fig. 3A), protopod with six coupling hooks and three plumose setae on medial margin, and three simple setae on lateral margin; rami with plumose setae; exopods slightly longer than endopod. Pleopod 2 (Fig. 3B), protopod rectangular, with five coupling hooks and eight plumose setae medially, one robust seta and four simple setae laterally; endopod smaller than exopod; appendix masculina inserted proximally, expanding distal end of endopod, reaching three-fourths length of endopod. Pleopods 3 and 4 (Figs. 3C and 3D), protopod with coupling hooks and plumose setae on medial margin and plumose setae on lateral margin; endopod much smaller than exopod, without plumose setae; exopod with plumose setae marginally and patch laterally; partial suture present on lateral margin. Pleopod 5 (Fig. 3E) subequal to pleopods 3 and 4, but endopod enlarged beyond protopod and without coupling hooks and plumose setae on medial margin.

Uropod (Figs. 1A, 1B and 3F), reaching distal end of pleotelson; protopod expanding distally on medial margin, one robust seta and eight simple setae on lateral margin; rami elongated oval, with numerous plumose and robust setae; endopod longer than exopod; apexes rounded.

Remarks. The material of R. excavata sp. nov. can easily be characterized as new to science by the following combinations of characters: (1) the rostrum is truncated anteriorly and stepped laterally; (2) eyes are separated from each other; (3) pereopods 1–3 have eight robust setae on the propodal blade and four or five blunt robust setae on each merus; (4) ischium to carpus in pereopods 4–7 have tubercles along the posterior margins; and (5) one pair of depressions is located at the proximal region of the pleotelson.

Among the known 41 species of the genus Rocinela, only three species have separated eyes and more than seven robust setae on the propodal blade in pereopods 1–3: R. niponia Richardson, 1909, R. garricki Hurley, 1957, and R. pakari Bruce, 2009 (Richardson, 1909; Bruce, 2009). Among them, Rocinela excavata sp. nov. most resembles R. garricki by sharing characteristics of the rostrum and propodal blade of pereopods 1–3. However, the former can be rapidly distinguished from the latter in terms of the distal end of the rostrum (truncated in the former vs. rounded in the latter) and the shape of the robust setae on the merus in pereopods 1–3 (blunt in the former vs. subacute in the latter). Rocinela excavata sp. nov. differs from the R. niponia and R. pakari in terms of the laterally stepped rostrum (vs. not stepped rostrum in the latter two species) and pereopods 4–7 having tubercles along the posterior margins (vs. smooth in the latter two species) (Bruce, 2009; Kim & Yoon, 2020).

Among seven species reported from the Far East, Rocinela excavata sp. nov. is most similar to R. japonica in the structure of rostrum and setal armature of pereopods 1–3’s merus, while the latter exhibits a distinct difference in the number of setae on the propodal blade in pereopods 1–3 (eight robust setae in the new species vs. three or four robust setae in R. japonica) (Richardson, 1898, 1904, 1909; Kussakin, 1974; Vasina, 1993). Rocinela excavata sp. nov. can be distinguishable from other six species by having separated eyes (vs. fused eyes in R. affinis), pereopod 1 bearing eight robust setae on the propodal blade (vs. less than eight in the latter six other species) (Richardson, 1904, 1909; Kussakin, 1979; Brusca & France, 1992).

Distribution. South Korea (Jeju Strait).

Prey (host). Unknown.

Etymology. The specific name, excavata, originates from the combination of Latin prefix ex- meaning “out of” and Latin word cavatus meaning “hollow out”. It refers to the shape of the rostrum laterally excavated; gender feminine.

Family Gnathiidae Leach, 1814

Genus Gnathia Leach, 1814

Type species. Gnathia termitoides Leach, 1814 (= Cancer maxillaris Montagu, 1804), by monotypy.

Diagnosis. Cephalon with generally straight frontal margin bearing frontal processes, while not deeply concaved; mandibles not elongated, with mandibular incisor and dentate blade; paraocular ornamentation and/or a dorsal sulcus present; pylopod distinct, 2 or 3-articled.

Remarks. Gnathiids show highly polymorphic forms depending on their developmental stages (Ota, 2014; Boxshall & Hayes, 2019). They are distinguishable from other cymothooids largely based on adult male’s characteristics of having remarkably enlarged mandibles and only five pairs of pereopods (Svavarsson & Bruce, 2012; Smit, Bruce & Hadfield, 2019). Among the gnathiids, the genera Anceus Risso, 1826, Praniza Latreille, 1817, and Zuphea Risso, 1826 have been traditionally regarded as junior synonyms of Gnathia, because they were based on gnathiid larval stages whose specific identifications cannot be possible (Cohen & Poore, 1994). Caecognathia Dollfus, 1901 and Elahpognathia Monod, 1926 have been elevated to generic rank by Cohen & Poore (1994). The genus Gnathia can be distinguished from others by its male characteristics such as a transverse frontal border on the cephalon having frontal processes, a 2- or 3-articled broad pylopod, and non-elongated mandibles having dentate blades (Cohen & Poore, 1994; Song & Min, 2018; Hadfield et al., 2019).

Gnathia obtusispina sp. nov.

urn:lsid:zoobank.org:act:3219C531-9A69-4805-B16D-5B14AF1B9B61.

Figures 4–6

Figure 4 Gnathia obtusispina sp. nov., holotype, male.

(A) Habitus, dorsal view; (B) Cephalon, dorsal view; (C) Cephalon, ventral view; (D) Cephalon, lateral view; (E) Mandible, lateral view; (F) Antennule; (G) Antenna; (H) Maxilliped; (I) Pylopod. Arrows indicate a tooth-like blunt spine. Scale bars: A = 1 mm, B–D = 0.5 mm, E–I = 0.2 mm.

Figure 5 Gnathia obtusispina sp. nov., holotype, male.

(A) Pereopod 2; (B) Pereopod 3; (C) Pereopod 4, (D) Pereopod 5; (E) Pereopod 6. Scale bar: A–E = 0.2 mm.

Figure 6 Gnathia obtusispina sp. nov., holotype, male.

(A) Pleopod 1; (B) Pleopod 2; (C) Pleopod 3; (D) Pleopod 4; (E) Pleopod 5; (F) Pleotelson and uropod. Scale bar: A–F = 0.5 mm.

Type material.—Holotype, designated here: South Korea: ♂, 3.2 mm, Hongdo-ri (34°43′22.8″N, 125°11′59.5″E), Heuksan-myeon, Sinan-gun, Jeollanam-do, 20 June 2018, 10 m depth, rinsing bryozoans and macroalgae on bedrock of sublittoral zones, S. H Kim, leg., SCUBA diving, NIBRIV0000900846. Paratypes: 2♂♂, same location as holotype, NIBRIV0000862802.

Description of holotype male. Body (Fig. 4A) 2.3 times longer than greatest width, with numerous long setae dorsally. Cephalon (Figs. 4B and 4C) oval to oblong, 0.4 times as long as wide, covered with numerous tubercles, with one pair of tooth-like paraocular ornamentations forming ridges (arrows in Figs. 4B and 4C); dorsal sulcus narrow, U-shaped, positioned at median area anteriorly; frontal border medially concave, with one pair of inferior frontolateral processes; frontal concavity shallow and narrow; supraocular lobes prominent, projecting upwards, with dentate apex; eyes located on lateral margins. Pereonites 1–4 covered with tubercles, whereas 5–7 without tubercles; pereonite 1 not fused to cephalon dorsally, immersed in posterior margin of cephalon; pereonites 2–4 subequal in length and width; pereonite 5 widest; pereonite 6 with concave posterior margin. Pleonites, epimera of pleonites 3–5 prominent. Pleotelson (Figs. 4A and 6F) triangular, with convex lateral margins; apex rounded, with three simple setae; proximal dorsal side with two pairs of simple setae and one pair of plumose setae.

Antennule (Fig. 4F), peduncular article 1 ovoid to oblong, with two penicillate setae laterally; article 2 square, 0.7 times longer than article 1, with four penicillate setae and three simple setae distally; article 3 elongate and rectangular, 1.5 times longer than article 2, with five simple setae distally; flagellar article 1 shortest, 0.1 times as long as peduncular article 3, with three penicillate setae laterally; article 2 elongated oblong, 4.9 times longer than article 1, with one simple seta and one aesthetasc distally; article 3 oblong, 0.3 times as long as article 2, with one aesthetasc distally; article 4 subequal to article 3 in length, with three simple setae and one aesthetasc distally. Antenna (Fig. 4G) peduncular article 1 globular; article 2 square, 0.8 times as long as article 1; article 3 oblong, 1.9 times longer than article 2, with two penicillate setae and seven simple setae distally; article 4 elongated rectangular, 1.3 times longer than article 3, with three penicillate setae and six simple setae distally; flagellum composed of five articles; each article square to oblong, similar each other in length, with simple setae distally.

Mandibles (Figs. 4A–4E) triangular, not elongate, half-length of cephalon, elevated distally, with dorsal and internal lobes; dentate blade irregular; basal neck indistinct. Maxilliped (Fig. 4H), endite reaching proximal region of palp article 2; palp articles globular, similar to each other in shape, article 1 with three plumose setae laterally; article 2 largest, with seven plumose setae laterally; article 3 with five plumose setae laterally; article 4 with seven plumose setae laterally and two short simple setae distally. Pylopod (Fig. 4I), article 1 longest, nearly occupying 70% of total length of pylopod, with numerous plumose setae on lateral margin, and 1 penicillate seta, one plumose seta, and 12 simple setae on medioventral side; article 2 ovoid, 0.2 times as long as article 1, with two short simple setae distally; article 1 elliptical, 0.3 times as long as article 2, with one simple seta on distal end.

Pereopod 2 (Fig. 5A) with tubercles on ischium to propodus inferiorly; basis with three penicillate setae superiorly, numerous simple setae superiorly and inferiorly; ischium 0.8 times as long as basis, with one serrate seta and six simple setae inferiorly, and one penicillate seta and four simple setae superiorly; merus 0.3 times as long as ischium, with one serrate seta and three simple setae inferiorly, and three simple setae superiorly; carpus similar to merus in length, with one simple seta and two serrate setae inferiorly; propodus oblong, 1.8 times longer than carpus, with two robust simple setae, one simple seta and several short simple setae on inferior margin, and one penicillate seta and one short simple seta at superodistal angle; dactylus rectangular, with four simple setae and one unguis distally. Pereopods 3–6 (Figs. 5B–5E) almost similar to pereopod 2; basis with tubercles superiorly except for pereopod 5.

Pleopods (Figs. 6A–6E) similar to each other; protopod ovoid to oblong, with one simple seta laterally, two coupling hooks medially; rami elongated ovoid, without plumose setae distally, except for pleopods 3 and 5; pleopod 2 with penicillate seta distally and one penicillate seta subdistally on exopod; pleopod 3 with three plumose setae on endopod and two plumose setae on exopod distally; pleopod 5 with one plumose seta on endopod and three plumose setae distally; appendix masculina not observed in pleopod 2.

Uropod (Fig. 6F), protopod rectangular, with one simple dorsal seta; rami with 7–10 simple setae along margin; endopod slightly longer than exopod, with 6–8 penicillate setae and 0–2 simple setae dorsally.

Remarks. In the 133 gnathia species, 13 species have paraocular ornamentations forming a ridge (Monod, 1926; Menzies, 1962; Schultz, 1966; Holdish & Harrison, 1980; Müller, 1993; Cohen & Poore, 1994; Pires, 1996; Tanaka, 2004; Kensley, Schotte & Poore, 2009; Ota, 2013; Song & Min, 2018; Shodipo et al., 2021). Among them, G. obtusispina sp. nov. most resembles two species, G. lignophila Müller, 1993 and G. andrei Pires, 1996, by having body integument covered by numerous tubercles (Müller, 1993; Pires, 1996). However, the new species can be easily distinguishable from these two species in that the frontal border of the cephalon is medially concave (vs. convex in the latter two species) and the pleotelson has rounded distal end (vs. acute distal end in the latter two species).

In the East Asia where the new species were collected, there are nine species characterized by the presence of tubercles on the cephalon and pereonites among 25 Gnathia species reported: G. tuberculata Richardson, 1909 from the Nanao, Japan; G. derzhavini Gurjanova, 1933 from the Askold Island, Russia; G. schmidti Gurjanova, 1933 from the Bay of Vladimir, Russia; G. teruyukiae Ota, 2011 from the Ishigaki Island, Japan; G. rufescens Ota, 2015 from the Okinawa Island Japan; G. albipalpebrata Ota, 2014 from the Okinawa-jima Island, Japan; G. parvirostata Ota, 2014 from the Ishigaki Island, Japan; G. nubila Ota & Hirose, 2009 from the Okinawa Island, Japan; and G. dejimagi Ota, 2014 from the Okinawa-jima Island, Japan (Boyko et al., 2008; Song & Min, 2018; Shodipo et al., 2021). Although G. obtusispina sp. nov. also represents this character state, this new species is easily distinguishable from the latter species by the combination of the following character states: (1) the body is covered with long setae; (2) the cephalon has a pair of remarkable tooth-like blunt paraocular ornamentations; (3) the frontal border of the cephalon is medially concave; (4) two inferior frontolateral processes are present ventrally; (5) the supraocular lobe is prominent and projecting upwards; (6) the dentate blade of the mandible is present and irregular; (7) pereonite 1 is not fused with cephalon dorsally and conspicuous; and (8) the apex of the pleotelson is rounded (Richardson, 1909; Gurjanova, 1933; Ota, 2011, 2015; Ota & Hirose, 2009).

Among the above-mentioned species, G. obtusispina sp. nov. is most similar to G. tuberculata by having inferior frontolateral processes and prominent supraocular lobes on cephalon, and mandible as long as half-length of the cephalon. However, the former differs from the latter in terms of the medially concave frontal border of the cephalon (vs. produced in the latter), presence of a tooth-like paraocular ornamentations (vs. absent in the latter), number of inferior frontolateral processes (two in the former vs. four in the latter), and rounded apex of the pleotelson (vs. acute in the latter) (Richardson, 1909).

Distribution. South Korea (the Yellow Sea)

Host. Unknown.

Etymology. The specific name, obtusispina, originates from the combination of Latin words obtusus, meaning “blunt” and spina, meaning “thorn”. This name refers to tooth-like paraocular ornamentation; gender feminine.

Discussion

Rocinela is distributed worldwide. It particularly shows high-latitude diversity (Bruce, 2009). Indeed, based on marine ecoregions of the world by Spalding et al. (2007), 29 of 41 known Rocinela species have been reported from a temperate region (Table 1). Among the temperate species, 21 known species are recorded from the Pacific, with 12 species from the temperate Northern Pacific region, including seven species from the Far East. This means that the majority of Rocinela species have been described from the temperate Northern Pacific, so the region could be considered as diversity hotspot for the genus Rocinela. However, given that Bruce (2009) has mentioned that a significant number of undescribed species from the tropical western Pacific region is held at the Muséum national d’Histoire naturelle in Paris, the lack of attention on the Rocinela species was likely to negatively affect our knowledge of the Rocinela species diversity in trophic region. So, undescribed species can be discovered through further study in this region. While among 29 species are known from the temperate region, only two species, R. angustata and R. belliceps, show a broad distribution ranging from the Northwest to Northeast Pacific despite most Rocinela species having endemic distribution ranges (Richardson, 1904, 1905, 1909; Kussakin, 1979; Brusca & France, 1992). Considering that host-association times is correlated with the distribution range and that Rocinela species can attach to the host temporally, these endemic distribution ranges of Rocinela species might be due to their feeding strategy with temporary ectoparasites attaching to fishes in their particular life history (Bruce, 2009; Smit, Bruce & Hadfield, 2019). Although hosts of R. angustata and R. belliceps remain unknown, broad distribution ranges of these two species could be related to their host’s distribution patterns (Smit, Bruce & Hadfield, 2019).

Table 1 Summary of Rocinela species from the temperate region.

Species	Location	Biogeographic realms	References	
R. affinis Richardson, 1904	Japan (Numazu)	TNP	Richardson (1904)	
R. americana Schioedte & Meinert, 1879	USA (Maine)	TNA	Schioedte & Meinert (1879); Kussakin (1979)	
R. angustata Richardson, 1904	USA (Bering Sea to Washington); Japan (Manazuru Zaki)	TNP	Richardson (1904); Brusca & France (1992)	
R. austeralis Schioedte & Meinert, 1879	Chile (Straits of Magellan)	TSA	Schioedte & Meinert (1879)	
R. belliceps (Stimpson, 1864)	USA (Alaska to California); Mexico (Clarion Island); Russia (Sea of Okhotsk)	TNP; TEP	Brusca & France (1992); Kussakin (1979)	
R. bonita Bruce, 2009	New Zealand (Bounty Trough)	TA	Bruce (2009)	
R. cornuta Richardson, 1898	USA (off Shumagin Bank)	TNP	Richardson (1898)	
R. danmoniensis Leach, 1818	Europe (Bay of Biscay to Iceland)	TNA	Bruce (2009)	
R. dumerilii (Lucas, 1849)	Mediterranean Sea	TNA	Bruce (2009)	
R. excavata sp. nov.	South Korea (Chujado Island)	TNP	Present study	
R. garricki Hurley, 1857	New Zealand (Cook strait)	TA	Hurley (1957)	
R. granulosa Barnard, 1914	South Africa (Natal)	TSAf	Barnard (1914b)	
R. Japonica Richardson, 1898	Japan (Hakodate Bay)	TNP	Richardson (1898)	
R. juvenalis Menzies & George, 1972	Peru (off Peru)	TSAm	Bruce (2009)	
R. kapala Bruce, 1988	Australia (New South Wales)	TA	Bruce (1988)	
R. laticauda Hansen, 1897	Mexico (off Acapulco); USA (California)	TEP; TNP	Brusca & France (1992)	
R. leptopus Bruce, 2009	New Zealand (Pagasus Bay)	TA	Bruce (2009)	
R. lukini Vasina, 1993	Sea of Okhotsk	TNP	Vasina (1993)	
R. maculata Schioedte & Meinert, 1879	Russia (Vladivostok)	TNP	Schioedte & Meinert (1879)	
R. niponia Richardson, 1909	Japan (Sado Island); South Korea (Chujado Island)	TNP	Richardson (1909); Kim & Yoon (2020)	
R. ophthalmica Milne Edwards, 1840	Italy (Sicily)	TNA	Bruce (2009)	
R. pakari Bruce, 2009	New Zealand (Chatham Rise)	TA	Bruce (2009)	
R. patriciae Brasil Lima, 1986	Brazil (off Rio Grande do Sul)	TSAm	Bruce (2009)	
R. propodialis Richardson, 1905	USA (Washington)	TNP	Richardson (1905)	
R. resima Bruce, 2009	New Zealand (Christabel Sea Mount)	TA	Bruce (2009)	
R. satagia Bruce, 2009	New Zealand (Chatham Rise)	TA	Bruce (2009)	
R. sila Hale, 1925	Australia (Adelaide)	TA	Hale (1925)	
R. tridens Hatch, 1947	USA (Washington)	TNP	Hatch (1947)	
R. tropica Brasil Lima, 1986	Brazil (Espírito Santo)	TSAm	Bruce (2009)	
R. tuberculosa Richardson, 1898	Mexico (Baja California)	TNP	Richardson (1898)	
Note:

TA, Temperate Australasia; TEP, Temperate Eastern Pacific; TNA, Temperate Northern Atlantic; TNP, Temperate Northern Pacific; TSAf, Temperate Southern Africa; TSAm, Temperate Southern America.

Fifty-six and 76 species of 133 known Gnathia species have been reported from a temperate region and tropical region, respectively (Table 2; Song & Min, 2018; Shodipo et al., 2021). Only two species, G. fragilis Schultz, 1977 and G. tuberculosa (Beddard, 1886), are from the Southern Ocean, Antarctic (Monod, 1926; Schultz, 1977). According to the marine ecoregions of the world, the Central Indo-Pacific (with 47 species) is thought to be the most diverse hotspot of Gnathia (Shodipo et al., 2021). After the Central Indo-Pacific, the second-most rich species of 18 species have been reported from the temperate Northern Pacific that includes the study area of the present study. Consequently, the temperate Northern Pacific is considered to be the second most diverse hotspot following the Central Indo-Pacific. Within the temperate Northern Pacific, the Far East, from which 11 Gnathia species are recorded, could be regarded as a representative hotspot. While looking for substrate types from which Gnathia species are collected, most temperate species have been collected from soft substrates such as mud, silt, and sandy flats in contrast to tropical Gnathia species reported from coral-reef habitats (Cohen & Poore, 1994; Svavarsson & Bruce, 2012). This result is a mismatch to the general knowledge that gnathiid species prefer coral reef-associated habits (Cohen & Poore, 1994; Santos & Sikkel, 2019; Smit, Bruce & Hadfield, 2019; Svavarsson & Bruce, 2012, 2019). Furthermore, the feature of the substratum strongly affects the distribution of gnathiids, and each species has a different habitat depending on its life stages (Smit, Bruce & Hadfield, 2019). Taken all together, the life history of Gnathia species is likely to differ depending on whether they live in a temperate or a tropic region (Santos & Sikkel, 2019). However, further study about the substratum preference between temperate and tropic Gnathia species is needed because most ecological studies of these species have been conducted from coral reef-associated habitats (Grutter, Morgan & Adlard, 2000, Grutter et al., 2018; Santos & Sikkel, 2019; Smit, Bruce & Hadfield, 2019; Shodipo et al., 2021). Additionally, although most Gnathia species are known as endemic, two species, Gnathia calmani Monod, 1926 and Gnathia nasuta Nunomura, 1992, have wide distributions ranging from the tropic to the temperate region (Monod, 1926; Holdish & Harrison, 1980; Nunomura, 1992; Ota, 2013). Another two species, G. grandilaris Coetzee et al., 2008 and G. trimaculata Coetzee et al., 2009, have been reported only from the Central Indo-Pacific, and also show a wide geographical distribution ranging from Australia to Japan (Coetzee et al., 2008, 2009; Ota & Hirose, 2009). According to Shodipo et al. (2021), the long-distance dispersal of some Gnathia species was facilitated by their host that had a wide movement radius in a short period of time (e.g., sharks). Considering wide movement radii of hosts such as sharks and rays in G. grandilaris and G. trimaculata, the two species showing wide distribution ranges, G. calmani and G. nasuta, also could be parasites of hosts having wide movement radii (Coetzee et al., 2008, 2009; Shodipo et al., 2021).

Table 2 Summary of Gnathia species from the temperate region.

Species	Location	Biogeographic realms	References	
G. africana Barnard, 1914	South Africa (Cape Town)	TSAf	Barnard (1914a); Monod (1926); Smit, As & Basson (1999); Smit, Van As & Basson (2002)	
G. albescens Hansen, 1916	Denmark (Foroe Island)	TNA	Hansen (1916)	
G. andrei Pires, 1996	Brazil (Ubatuba continental slope)	TSAm	Pires (1996)	
G. brachyuropus Monod, 1926	New Zealand (Akaroa, Lyttelton)	TA	Monod (1926)	
G. brucei George, 2003	USA (North Carolina)	TNA	George (2003)	
G. bungoensis Nunomura, 1982	Japan (Saeki Bay)	TNP	Nunomura (1982)	
G. calamitosa Monod, 1926	Australia (New South Wales)	TA	Monod (1926)	
G. calmani Monod, 1926	Australia (Heron Island; Victoria)	CIP; TA	Monod (1926)	
G. campontus Cohen & Poore, 1994	Australia (Bass Strait)	TA	Cohen & Poore (1994)	
G. capillata Nunomura & Honma, 2004	Japan (Sado Island)	TNP	Nunomura & Honma (2004)	
G. clementensis Schultz, 1966	USA (California)	TNP	Schultz (1966)	
G. coronadoensis Schultz, 1966	USA (Coronado canyon)	TNP	Schultz (1966)	
G. dentata (G. O. Sars, 1872)	Norway (Hardangerfijord)	TNA	Monod (1926)	
G. derzhavini Gurjanova, 1933	Russia (Askold Island)	TNP	Gurjanova (1933)	
G. disjuncta Barnard, 1920	South Africa (Cape Town)	TSAf	Monod (1926)	
G. epopstruma Cohen & Poore, 1994	Australia (Bass Strait)	TA	Cohen & Poore (1994)	
G. fallax Monod, 1926	Spain (Bay of Biscay)	TNA	Monod (1926)	
G. gurjanovae Golovan, 2006	Russia (Peter the Great Bay)	TNP	Golovan’ (2006)	
G. hirsuta Schultz, 1966	USA (California)	TNP	Schultz (1966)	
G. illepidus (Wagner, 1869)	Mediterranean Sea (Italy, Monaco)	TNA	Monod (1926)	
G. incana Menzies & George, 1972	Peru (off Peru)	TSAm	Menzies & George (1972); Cohen & Poore (1994)	
G. inopinata Monod, 1925	Mediterranean Sea (Italy, Monaco)	TNA	Monod (1926)	
G. iridomyrmex Cohen & Poore, 1994	Australia (Victoria)	TA	Cohen & Poore (1994)	
G. koreana Song & Min, 2018	South Korea (Geomundo Island)	TNP	Song & Min (2018)	
G. lacunacapitalis Menzies & George, 1972	Peru (off Peru)	TSAm	Menzies & George (1972); Cohen & Poore (1994)	
G. maxillaris (Montagu, 1804)	England (Cornwall)	TNA	Monod (1926)	
G. mulieraria Hale, 1924	Australia (Gulf St. Vincent)	TA	Hale (1924)	
G. mutsuensis Nunomura, 2004	Japan (Asamushi)	TNP	Nunomura (2004)	
G. mystrium Cohen & Poore, 1994	Australia (Bass Strait)	TA	Cohen & Poore (1994)	
G. nasuta Nunomura, 1992	Japan (off Tomioka; Amai; Keramal Okinawa islands)	CIP; TNP	Nunomura (1992)	
G. nkulu Smit & Van As, 2000	South Africa (Port Alfred)	TSAf	Smit & Van As (2000)	
G. notostigma Cohen & Poore, 1994	Australia (Bass Strait)	TA	Cohen & Poore (1994)	
G. obtusispina sp. nov.	South Korea (Hongdo Island)	TNP	Present study	
G. odontomachus Cohen & Poore, 1994	Australia (Victoria)	TA	Cohen & Poore (1994)	
G. oxyuraea (Lilljeborg, 1855)	North Sea	TNA	Monod (1926)	
G. panousei Daguerre de Hureaux, 1971	Morocco	TNA	Boyko et al. (2008)	
G. pantherina Smit & Basson, 2002	South Africa (Jeffreys Bay)	TSAf	Smit & Basson (2002)	
G. phallonajopsis Monod, 1925	Mediterranean Sea (France, Italy, Monaco, Sapin)	TNA	Monod (1926)	
G. pilosus Hadfield, Smit & Avenant-Oldewage, 2008	South Africa (Sheffield Beach, Tinley Manor)	TSAf	Hadfield, Smit & Avenant-Oldewage (2008)	
G. productatriedns Menzies & Barnard, 1959	USA (California)	TNP	Menzies & Barnard (1959)	
G. prolasius Cohen & Poore, 1994	Australia (Bass Strait)	TA	Cohen & Poore (1994)	
G. rectifrons Gurjanova, 1933	Russia (East Sea)	TNP	Gurjanova (1933)	
G. ricardoi Pires, 1996	Brazil (Ubatuba continental slope)	TSAm	Pires (1996)	
G. sanrikuensis Nunomura, 1998	Japan (Otsuchi Bay)	TNP	Nunomura (1998)	
G. schmidti Gurjanova, 1933	Russia (Bay of Vladimir)	TNP	Gurjanova (1933)	
G. serrulatifrons Monod, 1926	Mediterranean Sea	TNA	Monod (1926)	
G. sifae Svavarsson, 2006	New Zealand (Bay of Plenty)	TA	Svavarsson (2006)	
G. spongicola Barnard, 1920	South Africa (False Bay)	TSAf	Monod (1926)	
G. steveni Menzies, 1962	USA (California)	TNP	Menzies (1962)	
G. stigmacros Cohen & Poore, 1994	Australia (Bass Strait)	TA	Cohen & Poore (1994)	
G. teissieri Cals, 1972	Spain (Bay of Biscay)	TNA		
G. tridens Menzies & Barnard, 1959	USA (California)	TNP	Menzies & Barnard (1959)	
G. trilobata Schultz, 1966	USA (Coronado)	TNP	Schultz (1966)	
G. tuberculata Richardson, 1909	Japan (Nanoa)	TNP	Richardson (1909)	
G. ubatuba Pire, 1996	Brazil (Ubatuba continental slope)	TSAm	Pires (1996)	
G. venusta Monod, 1925	Mediterranean Sea (Monaco)	TNA	Monod (1926)	
G. vorax (Lucas, 1849)	Mediterranean Sea (Algeria, Bay of Biscay, Cape Bojador)	TNA	Monod (1926)	
Note:

CIP, Central Indo-Pacific; TA, Temperate Australasia; TEP, Temperate Eastern Pacific; TNA, Temperate Northern Atlantic; TNP, Temperate Northern Pacific; TSAf, Temperate Southern Africa; TSAm, Temperate Southern America.

Conclusion

The present study of Korean ectoparasitic isopods revealed high species diversity of Rocinela and Gnathia species in the temperate Northern Pacific region by the discovery of two new species, Rocinela excavata sp. nov. and Gnathia obtusispina sp. nov. The two new species are the species records for the 13th Rocinela species and the 19th Gnathia species in this region, respectively. Our investigation on the geographical distributions of known Rocinela and Gnathia species indicated that the temperate Northern Pacific has the most Rocinela species and the second most Gnathia species in the regional species richness of each genus. It also showed that even if both genera indicate great diversity in the western Pacific, Rocinela species reveal high-latitude diversity while Gnathia species represent low-latitude diversity, particularly in the Central Indo-Pacific region.

Additional Information and Declarations

Competing Interests

Author Contributions

Field Study Permissions

Data Availability

New Species Registration

The authors declare that they have no competing interests.

Sung Hoon Kim conceived and designed the experiments, performed the experiments, analyzed the data, prepared figures and/or tables, authored or reviewed drafts of the article, and approved the final draft.

Jong Guk Kim analyzed the data, prepared figures and/or tables, authored or reviewed drafts of the article, and approved the final draft.

Seong Myeong Yoon analyzed the data, prepared figures and/or tables, authored or reviewed drafts of the article, and approved the final draft.

The following information was supplied relating to field study approvals (i.e., approving body and any reference numbers):

Field experiments were approved by

Ministry of Environment (MOE) of the Republic of Korea (project number: NIBR201902204; NIBR202102204).

The following information was supplied regarding data availability:

The raw data are line drawings of two new species and the morphological character.

The following information was supplied regarding the registration of a newly described species:

Rocinela excavata species LSID: urn:lsid:zoobank.org:act:9A4CC86D-6930-4FC6-9FC9-DBF105A2B285

Gnathia obtusispina species LSID: urn:lsid:zoobank.org:act:3219C531-9A69-4805-B16D-5B14AF1B9B61

Publication LSID: urn:lsid:zoobank.org:pub:7A53937A-F2EB-49C7-B8DA-F0AA36241310.

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
