# Peer review of "Two new temporary ectoparasitic isopods (Cymothoida: Cymothooidea) from Korean waters with a note on geographical distributions of Rocinela Leach, 1818 and Gnathia Leach, 1814"

_PeerJ, doi:10.7717/peerj.14593_

## Round 0.1 · original submission · Major Revisions

Your data are interesting; I think that both the reviewers gave interesting comments and suggestions that you will be certainly able to address; I look forward to seeing the edited version of your paper!

·

Basic reporting

Authors created 2 new species of ectoparasitic isopods from Korea. Their drawings are nice, and the text is focused towards 2 new species of ectoparasites. However, it is not found in any hosts, hence ectoparasitic can not be considered. Some parasites are taken from plankton samples, likewise, this may be parasitic form, but not found from the host, until you find the host, it can be designated as free living.
Literature reference is good, sufficient information has been analysed and described. Figures are tables are done well. I have given my comments in the word file track mode, authors need to follow and should answer the queries.

Experimental design

Yes, it's done well.

Validity of the findings

Interesting, but still they are not found from the host, and hence can not be considered as an ectoparasite.
Holotype males are given in the description, but figure shows female, authors need to revise on this.

Additional comments

I have given my comments in tracked mode, authors need to follow it.

Reviewer 2 ·

Basic reporting

The paper was well composed, with clear drawings and detailed descriptions. In certain instances, the English language and grammar are problematic and the sentences need to be looked at.
There are instances where the reference choice is questionable. Some references do not seem to be appropriate for the specific sentence, and others seem too broad to be used for that particular sentence. The authors should look into this and maybe add some more specific references for their paper.

Experimental design

The methods were mostly done with sufficient detail. More information on how the actual isopods were collected would be helpful.

Validity of the findings

The authors have provided detailed drawings and descriptions of what appears to be two new species. The differences between the new and known species are in some cases a bit limited. Perhaps more differences could be added to solidify the uniqueness of the species. The authors discuss the geographical distributions of the species but do not compare to species from other geographical regions, after stating some species are known from different regions. It would be good to compare to similar species from around the world to make sure they do not have a cosmopolitan species.

Additional comments

The authors have put together a nice paper with two new species and I look forward to seeing the published paper.

Annotated reviews are not available for download in order to protect the identity of reviewers who chose to remain anonymous.

---

## Round 0.2 · Minor Revisions

I fully agree with the referee, I think that with minor but customary revisions the paper can be certainly accepted.

·

Basic reporting

no comment

Experimental design

It has done well.

Validity of the findings

Novel species findings, it's important.

Additional comments

Manuscript has been revised well, I have following concerns which need to be addressed.
title may revise like this: Two new temporary ectoparasitic isopods (Cymothoida:Cymothooidea).
I have attached the file with my comments, please follow it and revise. Some sentences are not clear, shown in yellow markings.

---

## Round 0.3 · accepted · Accept

Authors addressed all the referee suggestions, accordingly the paper can be accepted for publication.

·

Basic reporting

Clearly written, literature referred well.

Experimental design

It's good to see the taxonomical write up and drawings are excellent.

Validity of the findings

Interesting, new species are highly valuable.

Additional comments

It can be accepted.